# Structural Characterization and Functional Analysis of Mevalonate Kinase from *Tribolium castaneum* (Red Flour Beetle)

**DOI:** 10.3390/ijms25052552

**Published:** 2024-02-22

**Authors:** Haogang Zheng, Yuanyuan Yang, Ying Hu, Jiaxuan Shi, Qiaohui Li, Yuanqiang Wang, Qingyou Xia, Pengchao Guo

**Affiliations:** 1Integrative Science Center of Germplasm Creation in Western China (CHONGQING) Science City, Biological Science Research Center, Southwest University, Chongqing 400716, Chinaliqiaohui0226@163.com (Q.L.);; 2Chongqing Key Laboratory of Sericultural Science, Chongqing Engineering and Technology Research Center for Novel Silk Materials, Southwest University, Chongqing 400715, China; 3School of Pharmacy and Bioengineering, Chongqing University of Technology, Chongqing 400054, China; wangyqnn@cqut.edu.cn

**Keywords:** mevalonate pathway, enzymatic activity, juvenile hormone biosynthesis

## Abstract

Mevalonate kinase (MevK) is an important enzyme in the mevalonate pathway that catalyzes the phosphorylation of mevalonate into phosphomevalonate and is involved in juvenile hormone biosynthesis. Herein, we present a structure model of MevK from the red flour beetle *Tribolium castaneum* (*Tc*MevK), which adopts a compact α/β conformation that can be divided into two parts: an N-terminal domain and a C-terminal domain. A narrow, deep cavity accommodating the substrate and cofactor was observed at the junction between the two domains of *Tc*MevK. Computational simulation combined with site-directed mutagenesis and biochemical analyses allowed us to define the binding mode of *Tc*MevK to cofactors and substrates. Moreover, *Tc*MevK showed optimal enzyme activity at pH 8.0 and an optimal temperature of 40 °C for mevalonate as the substrate. The expression profiles and RNA interference of *TcMevK* indicated its critical role in controlling juvenile hormone biosynthesis, as well as its participation in the production of other terpenoids in *T. castaneum*. These findings improve our understanding of the structural and biochemical features of insect Mevk and provide a structural basis for the design of MevK inhibitors.

## 1. Introduction

The mevalonate pathway is a classic pathway in most living organisms that is involved in the synthesis of isopentenyl pyrophosphate, which is the building block for the synthesis of many biologically active compounds, including ubiquinone, cholesterol, steroid hormones, and isoprenoids [1,2]. The mevalonate pathway begins with the reductive polymerization of acetyl-CoA molecules to form 3-hydroxy-3-methylglutaryl-CoA, which is then reduced to mevalonate by 3-hydroxy-3-methylglutaryl-CoA reductase. Subsequently, mevalonate is phosphorylated to phosphomevalonate via catalysis by mevalonate kinase (MevK) and phosphomevalonate kinase. Mevalonate kinase transfers the γ-phosphoryl group from adenosine triphosphate (ATP) to the 5-hydroxyl oxygen group of mevalonate to produce phosphomevalonate. A second phosphoryl group is then added by phosphomevalonate kinase to produce 5-pyrophosphomevalonate, which is decarboxylated by mevalonate diphosphate decarboxylase to form isopentenyl pyrophosphate, the active precursor of isoprenoid [1,3]. Owing to their vital contribution to the biosynthesis of steroids and isoprenoids, enzymes in the mevalonate pathway are currently being explored as targets for antimicrobial and anticancer drugs [4,5]. In addition to the biosynthesis of diverse non-sterol isoprenoid metabolites with various cellular functions, insects also utilize the products of the mevalonate pathway to produce juvenile hormones (JHs) through the specific JH-branch pathway [1,6]. JHs refer to a family of lipophilic sesquiterpenoid molecules primarily synthesized in the corpora allata of insects, which regulate diverse physiological processes, such as development, metamorphosis, and reproduction in insects by changing their concentration at precise developmental times [7,8,9]. Thus, enzymes of the JH-branch and mevalonate pathways are considered targets for the development of new insect growth regulators or insecticides because they play an important role in insect development and reproduction by regulating JH synthesis [10,11,12].

A key regulatory step in the mevalonate pathway is the phosphorylation of mevalonate to produce phosphomevalonate, which is catalyzed by MevK, a member of the GHMP superfamily, which contains galactokinase, homoserine kinase, MevK, and phosphomevalonate kinase [13]. These kinases have low sequence identity but share highly conserved glycine-rich motifs for ATP binding. MevKs are widely found in eukaryotes, archaebacteria, and eubacteria. According to previous reports on MevK structures, all enzymes contain two domains, the N-terminal domain and the C-terminal domain, and the substrates bind within a cavity between the N- and C-terminal domains [14,15]. However, in mammals, this enzyme is a homodimer in solution with a subunit molecular weight of approximately 42 kDa [16,17], whereas the molecular weights of *Leishmania major* MevK and *Streptococcus pneumoniae* MevK are 37 kDa and 32 kDa, respectively [14,18]. Differences in the primary structure of mammalian and protozoan MevK may affect their three-dimensional structure and catalytic activity. To better understand changes in the structure and enzymatic activity of MevKs during evolution, further structural and activity characterization of MevKs is required. However, limited information is currently available on the structure of insect MevK. Insect MevKs are predominantly expressed in the *corpora allata* [19] and are involved in the mevalonate pathway. In silkworms, knocking down *MevK* through RNA interference caused females to lay diapause eggs [20].

The red flour beetle *Tribolium castaneum* is one of the most harmful pests in stored products and grains and has emerged as an important insect model system for a variety of topics. In this study, we conduct structural and enzymatic characterization of MevK from *T. castaneum* (*Tc*MevK). Structural and biochemical analyses were employed to define the binding mode of *Tc*MevK with cofactors and substrates. Moreover, we determined the expression profile and RNA interference of *Tc*MevK, which indicates that *Tc*MevK may not only be involved in JH biosynthesis but also relevant for the production of other terpenoids.

## 2. Results and Discussion

### 2.1. Structural Model of TcMevK

The structural model of *Tc*MevK was constructed using the ROBETTA server (https://robetta.bakerlab.org/). The quality of the models was evaluated using the CAMEO (continuously evaluating the accuracy and reliability of predictions), which yielded a confidence value of 0.82. The *Tc*MevK model was organized into two distinct domains: an N-terminal domain and a C-terminal domain. The N-terminal domain comprises a large five-stranded β-sheet (β1-β4-β7-β5-β6) and a small antiparallel β-sheet (formed by β2, β3, β8, and β9) extending upward from the large β-sheet. Only one side of the large β-sheet is packed with six α-helices (α1–α6), whereas the opposite side is exposed to the solvent (Figure 1A). The C-terminal domain contains another antiparallel β-sheet formed by four strands (β11-β12-β10-β13) and five long α-helices (α7–α11) covering the solvent side of the antiparallel β-sheet. A narrow and deep cavity exists at the junction between the two domains of *Tc*MevK, which most likely represents the binding site for the substrate and cofactor. To investigate the stability of the *Tc*MevK model, we used root mean square deviation (RMSD) to monitor the molecular dynamics simulation for 100 ns, and the RMSD plots showed the *Tc*MevK stabilized at ~2.2 Å after 50 ns (Appendix A).

Structural similarity analysis of the *Tc*MevK model was performed using the DALI server (http://ekhidna2.biocenter.helsinki.fi/dali/, accessed on 31 July 2023), which showed that *Tc*MevK is structurally similar to several MevKs, such as those in *Homo sapiens* (PDB ID:2R3V), *Rattus norvegicus* (PDB ID:1KVK), *Leishmania major* (PDB ID:2HFS), and *Methanosarcina mazei* (PDB ID:6MDE). Superposition of *Tc*MevK against the human and rat counterparts yielded an overall root mean square deviation in the range of 1.9–2.2 Å over approximately 280 Cα atoms, suggesting that *Tc*MevK and vertebrate MevK adopt a similar pattern (Figure 1B,C). Conformational differences were mainly observed in the α-helix regions of N- and C-terminal domains, which resulted in a wider substrate-binding cavity in vertebrate MevK than in *Tc*MevK (Appendix A). In addition, a comparison of *Tc*MevK to *L. major* MevK and *M.mazei* MevK yielded an overall root mean square deviation of 1.2 Å over approximately 180 Cα atoms. As shown in Figure 1D,E, *Tc*MevK and MevK from bacteria and protozoa adopt highly similar conformations of the C-terminal domain and substrate-binding pocket. Conversely, the N-terminal domain of bacterial and protozoan MevK contains several fewer α-helix regions than that of *Tc*MevK. Multiple sequence alignment also showed that the N-terminal domains of MevKs exhibited sequence diversity, whereas the C-terminal domain and elements around the substrate-binding cavity in the N-terminal domain were relatively conserved during evolution (Figure 2). Sequence differences in the MevK N-terminal domain are not only observed in the amino acid composition but also in the sequence length, which results in significant molecular weight differences between the MevKs of different species. Specifically, insect MevK possesses an extended N-terminal sequence, whereas the mammalian and plant MevKs exhibit a shorter N-terminal segment, while bacterial and protozoan MevKs feature the briefest N-terminal region (Figure 2).

### 2.2. Substrate- and Cofactor-Binding Site of TcMevK

To explore the substrate- and cofactor-binding site of *Tc*MevK, we individually docked the cofactor ATP and mevalonate (Mev) into the *Tc*MevK structure using the program HADDOCK 2.2 [23]. We first docked the ATP and magnesium ion to the *Tc*MevK structure. Among the 10 initial docking clusters, the cluster with the lowest energy satisfied the best interaction constraints. This particular cluster is considered the most reliable, as it exhibits an overall backbone RMSD of 0.5 ± 0.2 Å, a HADDOCK score of −76.4 ± 1.5, and a Z-score value of −1.8. In this model, the ATP molecule is located in a deeper groove, with the adenosine moiety anchored to the bottom of the pocket and the triphosphate moiety pointing to the center of the pocket (Figure 3A). In detail, the adenine ring is surrounded by the side chains of Pro109, Val147, and Pro257, and the adenosine ribose is stabilized by Gln113 via hydrogen bonding. The triphosphate moiety of ATP is mainly fixed by a glycine-rich motif between β6 and α4 (residues 148GAGTGSSA155), as well as residues of Lys11, His213, Asp220, and Thr260. The GXGXGXXX (S/T/A) pattern in the loop between strand β7 and helix α4 is well conserved in the majority of P-loop kinases/NMP kinases and normally represents a conserved ATP-binding motif [20,21]. The γ-phosphoryl group of ATP is located near the substrate Mev and fixed by Lys11 and Asp220, which are the likely catalytic residues of the GHMP family reaction [22]. Magnesium ion is located near the phosphate groups of ATP and coordinated by the β- and γ-phosphates of ATP and the carboxyl group of Ser145 (Figure 3A).

The substrate Mev was further docked into the *Tc*MevK-ATP structure model using the HADDOCK 2.2. Among the 12 output clusters, the cluster of relatively lower energy with eight numbers satisfied the best interaction restrains. The overall backbone RMSD of 0.4 ± 0.1 Å and the Z-score value of −1.9 indicated that the model is somewhat reliable. In this model, Mev is located at the entrance of the substrate-binding cavity, with the carboxyl group fitting into a relatively positively charged part of the pocket, which is stabilized by the main chain of Val21 and the side chain of Lys234 (Figure 3B). The hydrocarbon backbone was surrounded by Val21 and Val264 side chains, which formed hydrophobic interactions with Mev. The C5-hydroxyl group of the hydrocarbon tail is anchored to the bottom of the pocket, which is stabilized by the side chains of Lys11, Ser217, and Asp220 (Figure 3B). The C5-hydroxyl group of Mev is positioned in close proximity to the γ-phosphate of ATP, with a distance of 4.7 Å (Appendix A). This close proximity enables the deprotonated C5-hydroxyl group to potentially initiate an attack on the γ-phosphate group. Mutational analyses were performed, and the binding activity of various mutants was determined to verify the functional relevance of specific amino acid contacts within *Tc*MevK. The binding affinity of *Tc*MevK toward Mev, analyzed by fluorescence spectrophotometry, was 30.01 ± 0.93 μM (Table 1). The replacement of Lys11, Ser217, Lys234, and Val264 with Ala led to a reduction in binding affinity by approximately two- to three-fold compared to that of the wild-type enzyme; however, the mutation of Val21 and Asp220 with Ala resulted in a comparable affinity to the wild-type enzyme. These results indicate that Lys11, Ser217, Lys234, and Val264 directly participate in substrate stabilization, whereas other residues may be involved in substrate recognition. Moreover, multiple sequence alignments showed that the residues involved in substrate binding are conserved in MevKs (Figure 2), indicating that MevKs share a relatively conserved active site.

### 2.3. Effect of pH and Temperature on Enzymatic Activity of TcMevK

The reaction velocities of *Tc*MevK at various substrate concentrations were fit to the Michaelis–Menten equation, with *Km* and *Vmax* values for *Tc*MevK toward Mev of 32.71 ± 1.09 µM and 28.99 ± 1.26 μmol/min/mg enzyme, respectively (Figure 4A). The *Km* value of *Tc*MevK toward Mev is comparable to that of *Catharanthus roseus* MevK and human MevK, which are 35 µM and 24 µM, respectively [17,24], but much lower than that of rat MevK, which is 228 µM [25]. The *Vmax* value of *Tc*MevK toward Mev is comparable to that of *C. roseus* MevK, human MevK, and rat MevK, which are 49, 37, and 30 μmol/min/mg enzyme, respectively [17,24,25]. These results indicate that the enzymatic activity of MevK is primarily determined by its conserved substrate-binding site despite the significant differences in the sequence length of MevK among different species.

To evaluate the optimal pH for *Tc*MevK activity, we tested its activity in the pH range of 4.0–9.0. As shown in Figure 4B, *Tc*MevK showed optimal activity at pH 8.0; enzyme activity remained robust under more alkaline conditions (pH 9.0), exceeding 90% of that under optimal conditions. Under neutral conditions (pH 7.0), enzyme activity was maintained at approximately 70% of its optimal value. However, *Tc*MevK was sensitive to a decrease in pH, with its activity decreasing sharply under acidic conditions. Specifically, activity decreased by approximately 90% at pH 6.0 and almost disappeared under more acidic conditions (pH 4.0 and 5.0). The pH profile of enzyme activity indicated that *Tc*MevK is a pH-sensitive enzyme. The optimal pH value for *Tc*MevK is consistent with that of *Bombyx mori* phosphomevalonate kinase, which exhibits optimal activity at pH 8.0 and maintains relatively high activity under more alkaline conditions [26]. The effect of temperature on *Tc*MevK activity was investigated between 20 °C and 70 °C (Figure 4C). The resulting activity profile had an irregular bell shape, with the highest activity observed at 40 °C. In detail, the enzyme activity increased with temperatures from 20 to 40 °C, then abruptly declined with additional temperature increases. The enzyme activity of *Tc*MevK at 30 °C and 50 °C was more than 90% of the maximum activity, whereas the activity at 60 °C decreased to 15% of the maximum activity. These results indicate that *Tc*MevK exhibits relatively suitable thermal stability, with strong activity toward the substrate at 50 °C.

### 2.4. Structural Stability Analysis of TcMevK

The secondary structure of *Tc*MevK was determined using far-UV circular dichroism spectroscopy (Figure 5A). Two negative bands appeared at 208 nm and 222 nm, and two positive bands appeared at 192 nm and 195 nm, indicating that *Tc*MevK has a mixed α-helix and β-strand structure. The α-helix and β-strand contents were 25.6% and 33.1%, respectively. To explore the thermal stability of *Tc*MevK, its secondary structure was monitored by shifting its CD spectra. As shown in Figure 5A, the *Tc*MevK structure was relatively stable from 20 °C to 40 °C; however, upon increasing the temperature to greater than 50 °C, the α-helix (θ_208nm_ and θ_222nm_) structure was quickly induced to denature or aggregate (Figure 5B,C). The midpoint of the denaturation curve, which represents the melting temperature, was calculated to be 41.41 ± 0.28 °C and 45.02 ± 0.71 °C, respectively. The melting temperatures of denaturation/aggregation were very close to the temperature of enzymatic inactivation for *Tc*MevK, indicating that the catalytic activity of *Tc*MevK depends on the stability of its structure. In addition, the stability of *Tc*MevK at different pH values was evaluated using CD spectroscopy. As shown in Figure 5D, the characteristic peaks of α-helix (θ_208nm_ and θ_222nm_) showed little variation from pH 3.0 to pH 9.0, whereas an additional positive peak appeared at 195 nm or 200 nm when the pH value was less than 6.0, which represents the β-strand or β-turn structure. This result indicates that the overall structure of *Tc*MevK was relatively stable at different pH conditions, but the flexible part of the enzyme may undergo conformational change under acidic conditions, such as the β-turn. The observed effect of pH on structural stability could explain the phenomenon whereby the catalytic activity of *Tc*MevK remains high under medium and alkaline conditions but decreases under acidic conditions (Figure 5D).

### 2.5. Expression Profile of TcMevK and Its Physiological Activity in T. castaneum

The expression pattern of *TcMevK* during the eight key developmental stages of *T. castaneum* was determined at the transcript level from egg to adult stages. As shown in Figure 6A, RT-qPCR analysis revealed low expression of the *TcMevK* transcript in the early egg stage, then a rapid increase in expression, which was maintained at a high level from the late egg stage to the early larval stage. Subsequently, *TcMevK* transcript expression decreased to a relatively low level in the late larval stage, then increased steeply during the pupal stage. At the larval stage, although the expression of the *TcMevK* transcript was lower than that at the late pupal stage, it remained at a relatively high level. To examine the physiological function of *Tc*MevK during the larval stage in vivo, RNAi-mediated knockdown of *TcMevK* was performed by injecting *dsMevK* at the beginning of the third-instar larval stage. The knockdown efficiency of *TcMevK* was quantified at the transcript level three and five days after injection. Compared to the EGFP dsRNA-injected controls, expression of the *TcMevK* transcript decreased by 63% and 48% three and five days after *dsMevK* injection, respectively (Figure 6B). However, the *dsMevK*-injected larvae did not exhibit significant changes in growth or metamorphosis, likely because of the limited effectiveness of *MevK* interference.

Furthermore, we determined the expression of Krüppel homologs (*TcKr-h1*), an important JH-response gene; a change in the Krüppel homolog expression level represented a change in JH titer. As shown in Figure 6C, the *TcKr-h1* transcript was significantly suppressed three days after *dsMevK* injection, indicating that *Tc*MevK plays a critical role in the control of JH biosynthesis. However, the *TcKr-h1* transcripts declined to a lower level five days after *dsRNA* injection in the *dsMevK* interference and the *dsEGFP* control groups. This phenomenon indicates that the titer of physiological JH decreased during this period, which may have been caused by a decline in the rate-limiting enzyme for JH biosynthesis, juvenile hormone acid methyltransferase (JHAMT). Therefore, suppression of *TcMevK* did not lead to a further reduction in the *TcKr-h1* transcript. JHAMT has been identified as a key enzyme controlling the rate of JH biosynthesis in many insects, including *B. mori* [10,27] and *Drosophila melanogaster* [28]. The transcript level of *TcJHAMT* was high in the embryonic and early larval stages and low at the end of the final larval instar. During the pupal and subsequent adult developmental stages, the *TcJHAMT* transcript was almost undetectable [28]. This expression pattern of *TcJHAMT* was inconsistent with that of the *TcMevK* transcript, which was expressed at relatively low levels in the late larval stage but at high levels during the pupal and early adult stages (Figure 6A). This inconsistency also indicates that *Tc*MevK, a key enzyme of the mevalonate pathway, is not necessarily exclusive to the biosynthesis of JHs but is also relevant for the production of other terpenoids [29].

## 3. Conclusions

In this study, we present the structural model of MevK, an important enzyme in the mevalonate pathway, from *T. castaneum*. *Tc*MevK adopts a compact α/β conformation that comprises an N-terminal domain and a C-terminal domain. Structural and sequence analyses revealed that *Tc*MevK has a low sequence identity compared to homologous proteins but shares similar structural folds, especially in the C-terminal domain and substrate-binding pocket region. In addition, computational simulations and biochemical assays enabled us to define continuous substrate-binding sites that can accommodate cofactors and substrates. According to the enzymatic characterization of *Tc*MevK in vitro, maximum activity occurs at 40 °C and in an alkaline environment. CD spectra revealed that the optimal conditions for *Tc*MevK enzyme activity depend on the stability of its structure. Furthermore, RNA interference with *TcMevK* was performed in the third-instar larval stage of *T. castaneum*, whereby *TcMevK* knockdown significantly suppressed the expression of the JH-response gene *TcKr-h1*. These results indicated that *Tc*MevK plays a critical role in controlling JH biosynthesis. Structural and functional information on JH biosynthetic enzymes would facilitate the discovery of target-specific inhibitors, which are expected to be excellent candidates for insect growth regulators and insecticides. However, *TcMevK* knockdown did not significantly affect larval growth or metamorphosis, probably because of the limited effectiveness of *TcMevK* interference. Thus, further investigations are required to elucidate the in vivo enzymatic efficiency and physiological functions of MevK in insects.

## 4. Materials and Methods

### 4.1. Overexpression and Purification of TcMevK and Mutants

Overexpression of *Tc*MevK protein was performed as described in previous studies [30]. Briefly, the DNA fragments of *Tc*MevK were amplified by PCR and cloned into a pET28a-derived vector. These constructs with an N-terminal hexahistidine tag were transformed into *Escherichia coli* BL21 (DE3) (Novagen, Madison, WI, USA) strain. The overexpression of recombinant *Tc*MevK protein was induced with 0.2 mM isopropyl-β-D-thiogalactoside (IPTG) at 16 °C for 20 h. Cells were harvested using centrifugation at 6000× *g* for 30 min and resuspended in lysis buffer (20 mM Tris-HCl pH 8.0, 200 mM NaCl). After high-pressure homogenization. After the centrifugation at 12,000× *g* for 30 min, the supernatants were collected and loaded onto a HiTrap nickel-chelating column (GE Healthcare, San Francisco, CA, USA) equilibrated with binding buffer (20 mM Tris-HCl, pH 8.0, 200 mM NaCl). The *Tc*MevK protein was eluted with 350 mM imidazole buffer and loaded onto a HiLoad 16/60 Superdex 200 column (GE Healthcare, USA) equilibrated with 20 mM Tris-HCl pH 8.5 and 50 mM NaCl. The purity of protein was estimated using sodium dodecyl sulfate-polyacrylamide gel electrophoresis (SDS-PAGE). Mutant proteins were expressed and purified in the same manner as the wild-type protein.

### 4.2. Docking Simulation

The docking simulations were performed by the HADDOCK 2.2 (high-ambiguity-driven protein−protein docking) online software (https://alcazar.science.uu.nl/services/HADDOCK2.2/haddockserver-expert.html, accessed on 6 August 2023). In this study, we first docked the ATP and magnesium ion to the TcMevK structure and then docked the substrate Mev into the MevK-ATP structural model. For blind docking, a rigid docking protocol was considered with ambiguous interaction restraints. The residues of the GXGXGXXX motif in *Tc*MevK were used as the active site residues for cofactor ATP, and the amino acid corresponding to the interacting residues of the crystal structure 2HFU were used as the active site residues for substrate Mev in MevK. Passive residues were selected automatically as the exposed surface neighbors of the active residues. The docking follows three different stages: rigid-body energy minimization, semi-flexible refinement by simulated annealing in torsional angle space, and final refinement in Cartesian space, with or without explicit solvent. The final structures were clustered using the pairwise backbone RMSD at the interface. Upon cluster-structural analysis, the 5 lowest energy models were selected, and among these, the best one was characterized based on the lowest HADDOCK score, electrostatic energy, and Z-score.

### 4.3. Binding Affinity Measurements

As previously reported, the binding affinity of *Tc*MevK to its substrate mevalonate (Mev) was determined using a fluorescence spectrophotometer (Hitachi F7100, Hitachi, Japan) with a 1 cm optical path quartz cuvette. To begin the determination process, dilute the *Tc*MevK protein to an appropriate concentration (approximately 5 μM). Next, a quartz cuvette containing 2000 μL of protein sample was incubated for 2 min at 28 °C. The slit width was 5 mm for both excitation and emission. Considering the high content of tryptophan (0.3%) and tyrosine (2.5%) residues in *Tc*MevK, we used an excitation wavelength at 280 nm to measure the intrinsic fluorescence of the tryptophan side chain. To determine the binding ability of the wild-type protein and substrate, Mev at a final concentration of 0–300 μM was added to the protein sample, respectively. The reaction was started by adding substrate, and the fluorescence quenching data were recorded in the range of 290 to 450 nm. The binding affinities between the *Tc*MevK mutants and the substrate were determined using the same method. Each reaction was repeated three times, and the *KD* value was calculated using GraphPad Prism 5.0.

### 4.4. Enzymatic Activity Assay of TcMevK

The enzymatic activity of *Tc*MevK was measured by quantifying the amount of NADH consumption, as described in previous studies with some modifications [31]. To determine the effect of pH on *Tc*MevK activity, a 300 μL reaction containing 50 mM phosphate buffer solution with different pH values (pH 4.0–9.0) and 0.1 μg of purified protein was incubated for 5 min. After incubation, the reaction regents containing 16 mM MgCl_2_, 75 mM KCl, 5 mM ATP, 0.8 mM NADH, 5 mM phosphoenolpyruvate, 10 U pyruvate kinase,10 U lactate dehydrogenase and 1 mM Mev were added, and then the consumption of NADH (ε_NADH_ = 6220 M^−1^cm^−1^) was monitored at 340 nm with Synergy H4 spectrophotometer (Bio-Tek, Winooski, VT, USA). The effect of temperature on *Tc*MevK activity was determined using the same reaction buffer (50 mM PBS, pH 8.0) and the same reagents from 20 °C to 70 °C.

To determine kinetic parameters of *Tc*MevK for substrate Mev, 16 mM MgCl_2_, 75 mM KCl, 5 mM ATP, 0.8 mM NADH, 5 mM phosphoenolpyruvate, 10 U pyruvate kinase,10 U lactate dehydrogenase and 0.1 μg purified protein was added in a 300 μL reaction mixture, and the reaction was started by adding Mev at different concentrations (0–500 μM). The decrease in absorbance at 340 nm was monitored using a Synergy H4 spectrophotometer as above. Each reaction was repeated three times, and the kinetic parameters were calculated using GraphPad Prism 5.0.

### 4.5. Circular Dichroism Spectroscopy

Far-UV CD spectra were performed using an MOS-500 circular dichroism spectrometer (BioLogic, Grenoble, Isère, France) with standard procedures. *Tc*MevK was diluted into 0.05 mg/mL with 50 mM phosphate buffer and transferred into a 1 cm quartz cell for incaution at 25 °C. After 5 min incubation, the CD spectra of *Tc*MevK in the far-UV region that monitors the secondary structures of protein were detected from 190 nm to 250 nm. The thermal stability of *Tc*MevK was determined using the 50 mM phosphate buffer (pH 8.0) from 20 °C to 90 °C. Furthermore, we investigated the stability of the *Tc*MevK secondary structure over a pH range of 3.0 to 9.0. The mean residue ellipticities at 208 and 222 nm were utilized to characterize structural changes induced by temperature following baseline corrections.

### 4.6. Expression Analysis of TcMevK

The wild type of *T. castaneum* Georgia-1 strain was used in this study, which was raised in whole wheat flour containing 5% brewer’s yeast in a growth chamber at 29 ± 1 °C, 65% humidity. The samples to assess the developmental expression of *T. castaneum* were taken from the early egg (EE, 1 day old), late egg (LE, 3 days old), early larvae (EL, 1 day of larvae stage), late larvae (LL, 20 days of larvae stage), early pupae (EP, 1 day of pupae stage), late pupae (LP, 3 days of pupae stage), early adult (EA, 1 day of adult stage), and late adult (LA, 7 days of adult stage) were collected during the eight key developmental stages of *T. castaneum*. All samples were frozen in liquid nitrogen and stored at −80 °C for RNA isolation. RNAs were extracted using TRIZOL reagent (Thermo Scientific, Waltham, MA, USA), and then cDNA samples were obtained using M-MLV reverse transcriptase (Invitrogen, Waltham, MA, USA) at 42 °C and stored at −80 °C. The cDNA samples were normalized using *T. castaneum* ribosomal protein S3 (*TcRps3*, GenBank: CB335975) as an internal control. RT-qPCR primers were designed and presented in Appendix A. Three independent replicates were performed for RT-qPCR.

### 4.7. RNA Interference of TcMevK

The template for double-stranded RNA (dsRNA) synthesis was amplified to dsDNA by PCR according to the *TcMevK* sequence. The dsRNA was generated using Transcript Aid T7 High Yield Transcription Kit (Thermo Scientific, Waltham, MA, USA) according to the instructions. Each individual larva was injected with 200 ng dsRNA of *TcMevK* at the third-larval stage (about 15 days old), and 200 ng *EGFP* dsRNA was injected as control. RNAi efficiency was assessed by determining the transcript levels of the target genes three and five days after dsRNA injection. Three larvae were randomly selected for RNA extraction, and the RNA samples were employed in RT-qPCR.

## Figures and Tables

**Figure 1 ijms-25-02552-f001:**
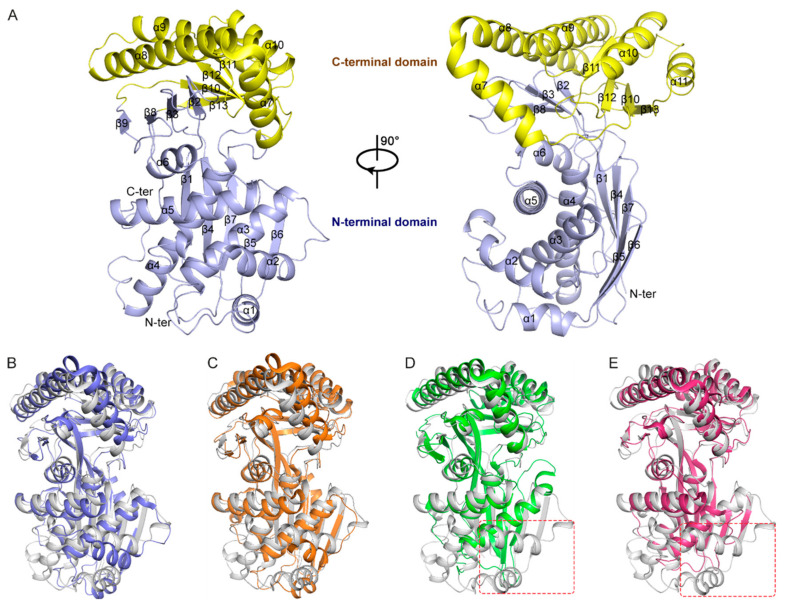
Structural analysis of *Tc*MevK. (**A**) Cartoon representation of the *Tc*MevK model in two orientations rotated by 90°. Light blue, N-terminal domain; yellow, C-terminal domain. Secondary elements are labeled. Comparison of the overall structure between *Tc*MevK (gray) and (**B**) *Homo sapiens* MevK (PDB ID: 2R3V, slate), (**C**) *Rattus norvegicus* MevK (PDB ID: 1KVK, orange), (**D**) *Leishmania major* MevK (PDB ID: 2HFS, green), and (**E**) *Methanosarcina mazei* MevK (PDB ID: 6MDE, warm pink). All figures were prepared using PyMOL2.5.5. The conformational diversity between *Tc*MevK and its counterparts is highlighted with a dashed frame.

**Figure 2 ijms-25-02552-f002:**
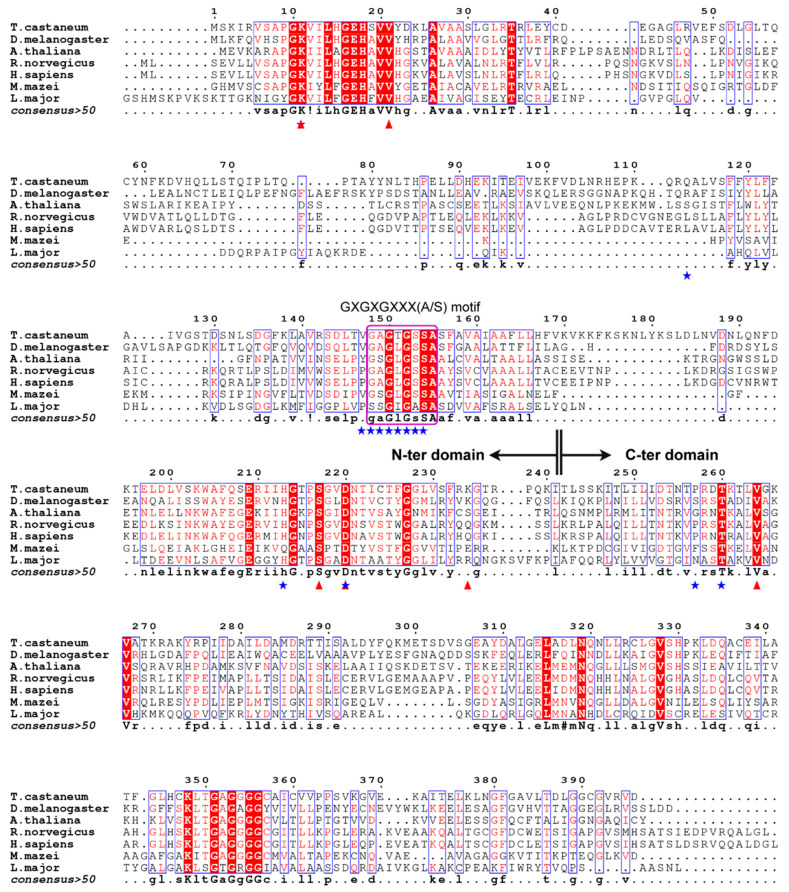
Multiple sequence alignment of *Tc*MevK and other MevK proteins show the conserved cofactor- and substrate-binding site. *T. castaneum*, *Tribolium castaneum* MevK (XP_972334.2); *D. melanogaster*, *Drosophila melanogaster* MevK (NP_001027412.1); *A. thaliana*, *Arabidopsis thaliana* MevK (NP_001190411.1); *R. norvegicus*, *Rattus norvegicus* MevK (XP_032742734.1); *H. sapiens*, *Homo sapiens* MevK (XP_047284829.1); *M. mazei*, *Methanosarcina mazei* MevK (WP_011033702.1); *L. major*, *Leishmania major* MevK (XP_001685041.1). The ATP-binding site is labeled as blue stars, and the substrate-binding residues are labeled as red triangles. The conserved GXGXGXX motif was highlighted with a purple frame. Alignments were performed with *ClustalW* [21] and *ESPript* [22].

**Figure 3 ijms-25-02552-f003:**
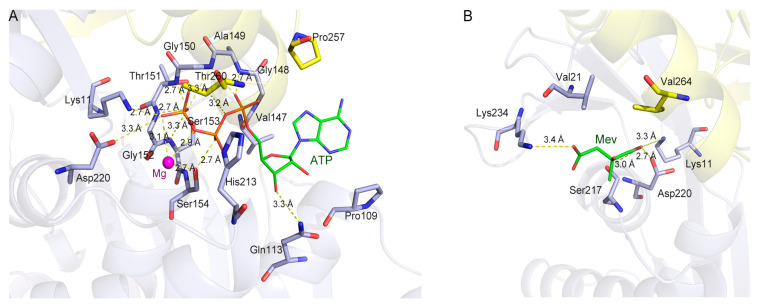
Substrate- and cofactor-binding sites of *Tc*MevK. (**A**) Binding pattern of cofactor ATP and Magnesium ion to the *Tc*MevK. (**B**) Binding pattern of substrate Mev to the *Tc*MevK. The residues involved in the cofactor and substrate binding are shown in stick format, docked Mev and ATP are green, and the magnesium ion is magenta. The hydrogen bonds and the salt bridges are yellow dashed lines. The backbone of protein is presented as a semitransparent cartoon.

**Figure 4 ijms-25-02552-f004:**
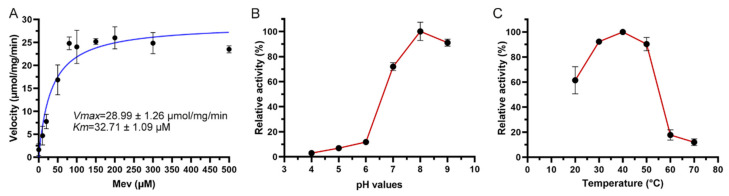
The enzymatic properties of *Tc*MevK toward Mev. (**A**) The kinetic parameters of *Tc*MevK toward Mev. (**B**) The effect of pH on the enzymatic activity of *Tc*MevK. (**C**) The curve diagrams of enzymatic activity of *Tc*MevK at different temperatures from 20 to 70 °C. Each point represents mean values from three independent tests.

**Figure 5 ijms-25-02552-f005:**
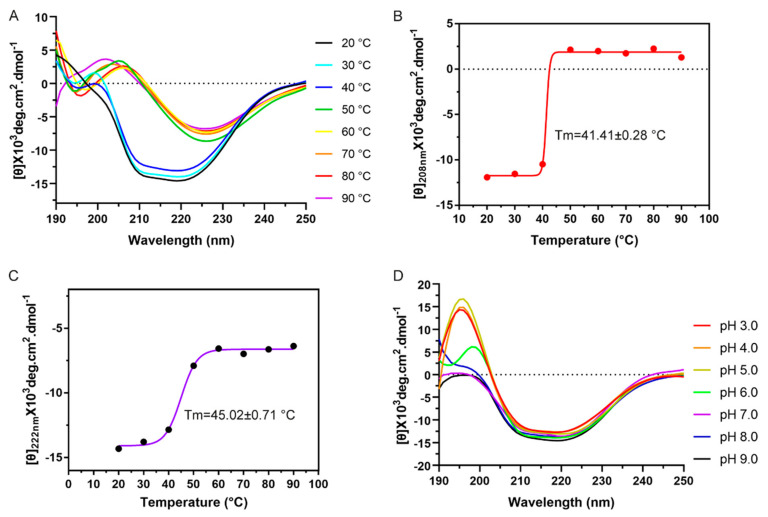
The secondary structure and thermodynamic analyses of *Tc*MevK. (**A**) The CD spectra of *Tc*MevK at different temperatures from 20 to 90 °C. Conformational changes of α-helix structure at 208 nm (**B**) and 222 nm (**C**) in *Tc*MevK induced by temperature. The mean residue ellipticities at 208 nm and 222 nm are used to monitor the conformational changes of *Tc*MevK induced by temperature. (**D**) The CD spectra of *Tc*MevK at different pH values from 3.0 to 9.0.

**Figure 6 ijms-25-02552-f006:**
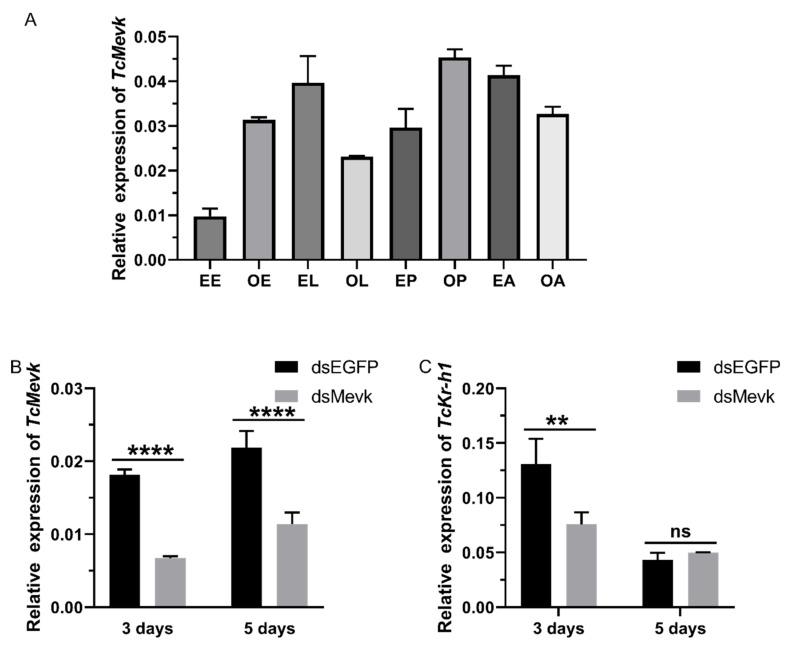
Expression patterns and RNA interference of *Tc*MevK in *T. castaneum*. (**A**) The expression patterns of *TcMevK* during the *T. castaneum* eight key developmental stages at the transcript level. EE, early egg; LE, late egg; EL, early larvae; LL, late larvae; EP, early pupae; LP, late pupae; EA, early adult; LA, late adult. (**B**) The knockdown efficiency of *TcMevK* was quantified three and five days after dsRNA injection. (**C**) The level of *Kr-h1* transcript was determined three and five days after *Tc*MevK knockdown. The vertical bars indicate standard errors of the mean (*n* = 3). Statistically significant differences from the control group are indicated with asterisks: **** *p* < 0.0001; ** *p* < 0.01; ns, no significance.

**Table 1 ijms-25-02552-t001:** The *KD* value of *Tc*MevK wild type and its mutants with Mev. Binding affinities were determined in 50 mM Tris-HCl, pH 8.0, and 50 mM NaCl at 25 °C (data represent mean ± s.d., *n* = 3 independent experiments).

	*KD* Value for Mev (μM)
*Tc*MevK	30.01 ± 0.93
K11A	75.80 ± 0.84
V21A	51.65 ± 2.16
S217A	70.15 ± 1.06
K234A	89.55 ± 2.45
V264A	57.14 ± 1.83
D220A	47.54 ± 2.17

## Data Availability

Dataset available on request from the authors.

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
