# Peer review of "Structural Characterization and Functional Analysis of Mevalonate Kinase from Tribolium castaneum (Red Flour Beetle)"

_ijms, 2024, doi:10.3390/ijms25052552_

Round 1

Reviewer 1 Report

Comments and Suggestions for Authors

This is an interesting paper on the modeling and experimental evaluation of the structure and binding affinity of mevalonate kinase from red flour beetle (TcMevK), which participates in the synthesis of its juvenile hormone that controls the growth of the larvae. The structure has been modeled with Robetta and compared with known homologs and substrate docking was studied by AutoDock. A putative binding site was identified and residues involved confirmed by mutation and substrate-binding experiments. The temperature-dependent CD experiments indicated a clear folding transition.

The paper is well written and easy to follow. The results should be useful in the design of TcMevK inhibitors as well as those of other MevKs. My only comment is that this research could benefit from checking the stability of the modeled structure and the modeled protein-substrate complex with molecular dynanmics, perhaps the authors consider it in further studies.  

Author Response

Dear reviewer:

Thank you very much for the thoughtful comments concerning our manuscript. We have addressed all questions and suggestions point by point.

Reviewer 1:

This is an interesting paper on the modeling and experimental evaluation of the structure and binding affinity of mevalonate kinase from red flour beetle (TcMevK), which participates in the synthesis of its juvenile hormone that controls the growth of the larvae. The structure has been modeled with Robetta and compared with known homologs and substrate docking was studied by AutoDock. A putative binding site was identified and residues involved confirmed by mutation and substrate-binding experiments. The temperature-dependent CD experiments indicated a clear folding transition.

  1. The paper is well written and easy to follow. The results should be useful in the design of TcMevK inhibitors as well as those of other MevKs. My only comment is that this research could benefit from checking the stability of the modeled structure and the modeled protein-substrate complex with molecular dynanmics, perhaps the authors consider it in further studies.

A1: Thanks for the suggestion. To investigate the stability of TcMevK model, we used root mean square deviation (RMSD) to monitor the molecular dynamics simulation for 100 ns, and the RMSD plots showed the TcMevK stabilized at ~2.2 Å after 50 ns (Fig. S1).

Reviewer 2 Report

Comments and Suggestions for Authors

The authors of the present work report the structural and functional characterization of an insect mevalonate kinase, TcMevK from the red floor beetle Tribolium castaneum. The authors construct a computational model of TcMevK, perform docking using HADDOCK, and identify amino acids implicated in substrate binding, which they confirm through mutational analysis. They also biochemically characterize the enzyme, by determining T and pH optimum, and assess the effect of pH and temperature on the secondary structure, also applying cd spectroscopy.

Last, they perform a transcriptomic analysis to monitor the expression levels of TcMevK gene during the various developmental phases and demonstrate its implication in juvenile hormone biosynthesis without though affecting larval growth or metamorphosis.

There are many points that need to be addressed:

1. In the introduction, the authors should also report (if any) previous studies on structure-function characteristics of insect mevalonate kinases.

2. The authors should present some statistics regarding the reliability of the modelled structure.

3. Figure S1: According to the manuscript, “A narrow and deep cavity exists at the junction between the two domains of TcMevK (Fig. S1)”, while in the figure legend the authors state that they show “Surface representation of the TcMevK dimer generated by symmetry operation. The substrate-binding pocket labeled as red rectangle box”. It is not clear whether the binding pocket is actually between the two domains of the monomer or between the two identical monomers that form the dimer. This should be clearly stated.

4. The statement  “Conformational differences were mainly observed in the α-helix regions of N- and C-terminal domains, which result in a wider substrate-binding cavity in vertebrate MevK than in TcMevK” is neither visually supported in Figure 1, nor by reporting distances in the main manuscript.

5. In figure 2, the distances of displayed bonds should be indicated.

6. Line 152: Asp is not a basic residue.

7. The authors should also state whether the distances of the two molecules (mevalonate and ATP) docked in the enzyme active site is close to the experimentally determined ones, if there are any, or in accordance with the anticipated reaction between them.

8. Line 340: it is not the thermal stability of enzyme activity being determined, it is the effect of temperature on enzyme activity.

Comments on the Quality of English Language

The language should be checked, as there are some minor errors throughout the manuscript eg in the phrase at line 291 "These results indicated..." "indicating" should be removed etc

Author Response

Dear reviewer:

Thank you very much for the thoughtful comments concerning our manuscript. We have addressed all questions and suggestions point by point.

The authors of the present work report the structural and functional characterization of an insect mevalonate kinase, TcMevK from the red floor beetle Tribolium castaneum. The authors construct a computational model of TcMevK, perform docking using HADDOCK, and identify amino acids implicated in substrate binding, which they confirm through mutational analysis. They also biochemically characterize the enzyme, by determining T and pH optimum, and assess the effect of pH and temperature on the secondary structure, also applying cd spectroscopy.

Last, they perform a transcriptomic analysis to monitor the expression levels of TcMevK gene during the various developmental phases and demonstrate its implication in juvenile hormone biosynthesis without though affecting larval growth or metamorphosis.

There are many points that need to be addressed:

  1. In the introduction, the authors should also report (if any) previous studies on structure-function characteristics of insect mevalonate kinases.

A1: Currently, there is a lack of information regarding the structure of insect Mevk. Therefore, in the introduction, we have incorporated studies focusing on the functional characteristics of insect Mevk.

  1. The authors should present some statistics regarding the reliability of the modelled structure.

A2: The parameters of the overall backbone RMSD value, the HADDOCK score, and the Z-score value of the docking model were added in the manuscript.

  1. Figure S1: According to the manuscript, “A narrow and deep cavity exists at the junction between the two domains of TcMevK (Fig. S1)”, while in the figure legend the authors state that they show “Surface representation of the TcMevK dimer generated by symmetry operation. The substrate-binding pocket labeled as red rectangle box”. It is not clear whether the binding pocket is actually between the two domains of the monomer or between the two identical monomers that form the dimer. This should be clearly stated.

A3: This sentence has been revised: “Surface representation of the TcMevK (A), Homo sapiens MevK (B), and Rattus norvegicus MevK (C) to show the substrate-binding pocket……...”

  1. The statement  “Conformational differences were mainly observed in the α-helix regions of N- and C-terminal domains, which result in a wider substrate-binding cavity in vertebrate MevK than in TcMevK” is neither visually supported in Figure 1, nor by reporting distances in the main manuscript.

A4: The substrate-binding cavity between TcMevk and vertebrate MevKs were compared and added into the revised Figure S1.

  1. In figure 2, the distances of displayed bonds should be indicated.

A5:The bonds distances were added in the revised figure.

  1. Line 152: Asp is not a basic residue.

A6: This sentence has been revised as: …as well as residues of Lys11, His213, Asp220, and Thr260…

  1. The authors should also state whether the distances of the two molecules (mevalonate and ATP) docked in the enzyme active site is close to the experimentally determined ones, if there are any, or in accordance with the anticipated reaction between them.

A7: Thanks, the distance of mevalonate and ATP was mentioned in the revised manuscript: ……The C5-hydroxyl group of Mev is positioned in close proximity to the γ-phosphate of ATP, with a distance of 4.7 Å (Fig. S2). This close proximity enables the deprotonated C5-hydroxyl group to potentially initiate an attack on the γ-phosphate group.

  1. Line 340: it is not the thermal stability of enzyme activity being determined, it is the effect of temperature on enzyme activity.

A8: Corrected as reviewer’s suggestion.

Comments on the Quality of English Language

The language should be checked, as there are some minor errors throughout the manuscript eg in the phrase at line 291 "These results indicated..." "indicating" should be removed etc

A9: The manuscript was carefully revised by a native speaker.

Reviewer 3 Report

Comments and Suggestions for Authors

Regarding the docking calculations, several key aspects necessitate clarification:

·      Was a blind docking procedure employed? Insight into the methodology used for identifying binding sites is crucial for a comprehensive understanding.

·      Considering that docking outcomes may be influenced by the presence of a co-factor, it is essential to determine which ligand entered the pocket first. Did the models contain the ATP or the substrate molecule?

·      How was the docking protocol validated? A standard practice involves utilizing a reference ligand, such as the X-ray ligand, to assess the tool's efficacy in reproducing the experimental binding mode and key interactions

·      The term 'reliable cluster model' at line 131 requires elucidation. Additional information supporting this statement should be incorporated.

·      Did the authors account for all protonation states of His213? This is pertinent, especially considering the existence of a protonated form of this residue.

·      information about the reaction catalyzed by the enzyme should be included. Does it involve ions? If so, this aspect must be incorporated into the model.

·      A dedicated section regarding docking should be included in the 'Materials and Methods' to provide a detailed account of the procedures followed in the docking analysis.

In the abstract, the sentence ‘Computational simulation’ is incorrect and should be changed in ‘docking simulation’

Author Response

Dear reviewer:

Thank you very much for the thoughtful comments concerning our manuscript. We have addressed all questions and suggestions point by point.

Regarding the docking calculations, several key aspects necessitate clarification:

  1. Was a blind docking procedure employed? Insight into the methodology used for identifying binding sites is crucial for a comprehensive understanding.

A1: Thanks, the HADDOCK 2.2 (high ambiguity driven protein−protein docking) software was employed, the docking protocol was added in the section of “4.2. Docking simulation”.

  1. Considering that docking outcomes may be influenced by the presence of a co-factor, it is essential to determine which ligand entered the pocket first. Did the models contain the ATP or the substrate molecule?

A: We firstly docked the ATP into the MevK structure model, and then Mev molecule was further docked into the MevK-ATP structure model.

  1. How was the docking protocol validated? A standard practice involves utilizing a reference ligand, such as the X-ray ligand, to assess the tool's efficacy in reproducing the experimental binding mode and key interactions

A3: In this study, we validated the docking model of MevK-Mev through rational mutagenesis and biochemical assays. However, we did not conduct point mutations or binding activity verification to validate the docking model of MevK-ATP. This was due to the extensive number of amino acids combined with ATP.

  1. The term 'reliable cluster model' at line 131 requires elucidation. Additional information supporting this statement should be incorporated.

A4: The term 'reliable cluster model' was elucidated and the relevant sentence was revised as: “Among the 10 initial docking clusters, the cluster with the lowest energy that satisfied the best interaction constraints. This cluster is the most reliable with the overall backbone RMSD of 0.6 ± 0.2 Å, the HADDOCK score of -76.4 ± 1.5, and the Z-Score value of -1.8……  

  1. Did the authors account for all protonation states of His213? This is pertinent, especially considering the existence of a protonated form of this residue.

A5: In this docking, we used the HADDOCK 2.2 online software, and selected the “automatically guess histidine protonation states using molprobity” option during the docking process. Thus, we did not account the protonation states of His213.

  1. Information about the reaction catalyzed by the enzyme should be included. Does it involve ions? If so, this aspect must be incorporated into the model.

A6: Yes, the MevK activity was dependent on the presence of divalent magnesium ions. The magnesium ion was docked into the structure and presented in the Figure. 3A.

  1. A dedicated section regarding docking should be included in the 'Materials and Methods' to provide a detailed account of the procedures followed in the docking analysis.

A7: Thanks for the suggestion, the docking protocol has been added in the section of “4.2. Docking simulation

  1. In the abstract, the sentence ‘Computational simulation’ is incorrect and should be changed in ‘docking simulation’.

A8: Thanks, it was corrected.

Round 2

Reviewer 2 Report

Comments and Suggestions for Authors

Thank you for answering the raised points. Regarding point (2), I was referring to the quality of the modelled structure using Robetta, not of the docking simulations. Please also include these statistics(ROBETTA scores). Also, in the "Methods" section, regarding the docking simulation, the authors should include Mg in sentence in line 338.

Author Response

Q1: Thank you for answering the raised points. Regarding point (2), I was referring to the quality of the modelled structure using Robetta, not of the docking simulations. Please also include these statistics(ROBETTA scores). Also, in the "Methods" section, regarding the docking simulation, the authors should include Mg in sentence in line 338.

A1: Thanks for the suggestion, the evaluation of the modelled structure was added and the sentence has been revised as: “The quality of the models was evaluated using the CAMEO (Continuously evaluate the accuracy and reliability of predictions), which yielded a confidence value of 0.82.” In addition, the magnesium ion was added in the line 338 of revised manuscript.